# Assessment of Landslide Susceptibility Using Integrated Ensemble Fractal Dimension with Kernel Logistic Regression Model

**DOI:** 10.3390/e21020218

**Published:** 2019-02-24

**Authors:** Tingyu Zhang, Ling Han, Jichang Han, Xian Li, Heng Zhang, Hao Wang

**Affiliations:** 1School of Earth Science and Resources, Chang’an University, Key Laboratory of Degraded and Unutilized Land Remediation Engineering, Ministry of Land and Resources, Shaanxi Provincial Key Laboratory of Land Rehabilitation, Xi’an 710064, China; 2Shaanxi Provincial Land Engineering Construction Group Co. Ltd., Xi’an 710075, China; 3School of Geological and Surveying & Mapping Engineering, Chang’an University, Xi’an 710064, China

**Keywords:** GIS, landslide susceptibility, fractal dimension, classification model

## Abstract

The main aim of this study was to compare and evaluate the performance of fractal dimension as input data in the landslide susceptibility mapping of the Baota District, Yan’an City, China. First, a total of 632 points, including 316 landslide points and 316 non-landslide points, were located in the landslide inventory map. All points were divided into two parts according to the ratio of 70%:30%, with 70% (442) of the points used as the training dataset to train the models, and the remaining, namely the validation dataset, applied for validation. Second, 13 predisposing factors, including slope aspect, slope angle, altitude, lithology, mean annual precipitation (MAP), distance to rivers, distance to faults, distance to roads, normalized differential vegetation index (NDVI), topographic wetness index (TWI), plan curvature, profile curvature, and terrain roughness index (TRI), were selected. Then, the original numerical data, box-counting dimension, and correlation dimension corresponding to each predisposing factor were calculated to generate the input data and build three classification models, namely the kernel logistic regression model (KLR), kernel logistic regression based on box-counting dimension model (KLR_box-counting_), and the kernel logistic regression based on correlation dimension model (KLR_correlation_). Next, the statistical indexes and the receiver operating characteristic (ROC) curve were employed to evaluate the models’ performance. Finally, the KLR_correlation_ model had the highest area under the curve (AUC) values of 0.8984 and 0.9224, obtained by the training and validation datasets, respectively, indicating that the fractal dimension can be used as the input data for landslide susceptibility mapping with a better effect.

## 1. Introduction

Landslides are regarded as one of the most destructive and frequently occurring natural disasters in the world. Globally, landslides cause about 1200 deaths and 3.5 billion dollars of loss each year [1]. China is a high-incidence region for landslides. Every year, it is reported that around 8935 landslides occur in China and about 350 people lose their lives due to landslides. Due to the diversity of the geological environment, the vagaries of climate, and the uneven distribution of the population, the spatial distribution of landslide risk in China is not uniform, which increases the obstructions in landslide control [2]. 

Landslide susceptibility mapping is one of the preliminary steps used to predict landslide occurrence, the main purpose of which is to divide a specified region into multiple classes that range from stable to unstable [3]. However, as the basic method to find landslide locations, field surveys are time consuming and have no predictive ability. With the development of geographic information systems (GISs), some statistical approaches, including bivariate and multivariate statistical methods, such as frequency ratio [4,5,6,7], index of entropy [8,9], certainty factors [10,11,12], statistical index [13,14], weights of evidence [15,16,17], analytic hierarchy process [18,19], fuzzy approaches [20,21], logistic regression [22,23], and evidential belief function [24,25], have widely been used to produce landslide susceptibility maps (LSMs) and can be seen in many studies. To obtain more accurate LSMs, various data mining algorithms have been applied in landslide susceptibility assessment, for example, support vector machine [26,27], decision trees [28,29], artificial neural network [30,31], adaptive neuro-fuzzy inference systems [32], multivariate adaptive regression spline [33,34,35], random forest [36,37,38], naive Bayes [39,40,41], naive Bayes trees [42], and kernel logistic regression [43]. 

In addition, the fractal theory for landslide susceptibility assessment can be seen in a few studies [44,45], but most of these have concentrated on the correlation between landslide distribution and fractal dimension. At present, there is basically no research on combining fractal dimension and data mining. Therefore, the aim of this study was to integrate two different types of fractal dimension to run the two-class kernel logistic regression to generate new hybrid models for landslide susceptibility mapping, namely the kernel logistic regression based on box-counting dimension model (KLR_box-counting_) and the kernel logistic regression based on correlation dimension model (KLR_correlation_), and compare these hybrid models with their archetypes in Baota District, Yan’an City, China.

## 2. Description of the Study Area

Baota District was selected as the study area and is located in the middle of a loess area, in Yan’an City, China. The geographical coordinates of the study area are between the 109°14′–110°07′ west–east longitudes and the 36°11′–37°02′ north–south latitudes. The study area is 96 km long, from north to south, and 76 km wide, from East to West, and covers an area of 3546 km^2^. 

The overall topography of the district presents a state of high in the east and low in the west, and a central uplift where the highest and lowest altitudes are 1464 and 860 m, respectively. The study area is located on the west side of the Yellow River Basin, with two major tributaries of the Yellow River, namely the Yan River and Fen Chuan River, and the annual average runoff of the Yan River is 2.93 × 10^8^ m^3^. The climate type of the study area belongs to a semi-humid semi-arid continental monsoon climate, the annual average temperature ranges from 7.7~10.6 °C, the average annual rainfall is approximately 540 mm, and most of the precipitation is concentrated in August.

The main lithologies are loess, sandstone, and mudstone. Around 17 geological units are distributed in the study region (Table 1). In addition, the neotectonic movement in the study area presents the intermittent uplifting movement of the crust and the undercut of the river, which generated the typical loess plateau landform. The data records showed that the deformation rate of the crust in the study area was 1 to 2 mm/a, and no earthquake with a magnitude 4 or above had occurred; therefore, earthquake-induced landslides were excluded in this paper.

## 3. Methodology

To build the landslide susceptibility model and obtain the LSM, there were four main steps in the present research: (1) Data preparation, including landslide inventory and a description of landslide predisposing factors; (2) landslide predisposing factor analysis, based on a series of indexes and methods; (3) landslide modeling using the KLR model, the KLR_box-counting_ model, and the KLR_correlation_ model; and (4) the models’ performance evaluation.

### 3.1. Data Preparation

#### 3.1.1. Landslide Inventory

The landslide inventory map, which reflects the relationship between predisposing factors and landslide distribution, is considered as the most crucial and essential phase before landslide susceptibility modeling [46]. Generally, it can obtain an inventory of the landslide location, category, occurrence date, size, volume, and active state [47]. In this study, the landslide inventory map was produced using existing literature and reports, field survey data, and the results from the interpretation of aerial photographs (Figure 1). There were 316 landslides including four debris flows, 295 rainfall-induced slides, and 17 falls in the landslide inventory map [48], and the largest plane proportion of landslides was approximately 11.4 × 10^4^ m^2^, the minimum area was about 295 m^2^, and the average proportion was 61 m^2^. The centroid method was applied to convert these landslide pattern spots into points to represent landslide locations. For subsequent landslide susceptibility modeling, the same number of non-landslides locations were randomly generated on the landslide inventory map. Then, a total of 632 points were divided into two parts according to the ratio of 70%:30%, with 70% (442) of the points used as the training dataset to train the models, and the remaining, namely the validation dataset, were applied for validation.

#### 3.1.2. Landslide Predisposing Factors

The reasons behind the causes of landslide occurrence are complicated; so far, there have been no consistent comments with regard to the determination of landslide predisposing factors. In this case study, thirteen types of landslide predisposing factors, including slope aspect, slope angle, altitude, lithology, mean annual precipitation (MAP), distance to rivers, distance to faults, distance to roads, normalized differential vegetation index (NDVI), topographic wetness index (TWI), plan curvature, profile curvature, and terrain roughness index (TRI), were employed according to observations in the wild and previous studies on the study area [49]. In addition, a 30 m-resolution digital elevation model (DEM) was used to extract the slope aspect, slope angle, altitude, TWI, TRI, and the plan curvature and profile curvature layers using ArcGIS tools. The lithology and MAP layers were produced based on 1:100,000 geological map and meteorological data collected from the local government. The GF-2 remote sensing image and the 1:50,000 topographical map were applied to construct the distance to rivers, distance to faults, distance to roads, and NDVI layers.

Slope aspect is a significant factor for slope stability and landslide distribution [50]. Different slope aspects receive different light radiation, which influences the water content of the soil. In this study, slope aspect was classified into nine directions using the natural break method as follows: flat, north, northeast, east, southeast, south, southwest, west, and northwest, respectively (Figure 2a).

In general, the probability of landslide occurrence increases with the increase of slope angle, which may influence the slope shear stress, and is still considered as one of the essential landslide predisposing factors by many scholars [51]. In this study, slope angle was classified into five sections using the natural break method as follows: 0–10.4469°; 10.4469–18.6711°; 18.6711–25.7839°; 25.7839–33.3412°; and 33.3412–56.4579°, respectively (Figure 2b).

Altitude is also an important predisposing factor for landslide occurrence [52]. The change of altitude affects the magnitude of slope stress and affects the potential energy of the landslide. Using the natural break method, the altitude value in the study area was classified into five ranges as follows: 848–1037.6823; 1037.6823–1128.4000; 1128.4000–1210.8706; 1210.8706–1298.8392; and 1298.8392–1549 m, respectively (Figure 2c).

Lithology is considered as the material basis of landslide development and occurrence. The weathering resistance and strength of rock and soil are determined by the types of lithology. On the other hand, the type and feature of landslides differ depending on the combination of rock mass with different properties, hardness, and structure [53]. According to geologic ages and lithofacies (Table 1), all of the geological units were reclassified into eight categories (Figure 2d).

Tectonic movement is not only one of the important factors in evaluating the regional geological stability, but is also a pivotal factor in landslide occurrence [54]. For this study, the value of distance to faults was employed to quantify the impact of faults on landslide occurrence and was reclassified into five ranges as follows: 0–2000; 2000–4000; 4000–6000; 6000–8000; and >8000 m, respectively (Figure 2e). 

River erosion plays a key role in the development of landslides. Many scholars believe that the effect of erosion on landslide stability is mainly reflected in the weakening of resistance of the landslide front and the increase of the free surface [55]. Therefore, the value of distance to rivers was employed to quantify the impact of river erosion on landslide development and was reclassified into five ranges according to the field observations and local conditions as follows: 0–200; 200–400; 400–600; 600–800; and >800 m, respectively (Figure 2f). 

Human activity is a primary factor that triggers landslides, as road construction is mainly the performance of human activities. The excavation of the slope and the earthwork accumulation during the construction process changes the local geological environment, which will directly or indirectly trigger a landslide [56]. In this study, the value of distance to roads was used as one of the condition factors and reclassified into five ranges: 0–200, 200–400; 400–600; 600–800; and >800 m, respectively (Figure 2g).

Rainfall is considered to be an important factor in landslides because the study area is covered by a large area of loess and the structure will become loose after the loess is immersed in water [57]. In this study, the value of MAP was employed to represent the influence of rainfall on landslides. The MAP was divided into six sections according to the intervals of 20 mm/yr as follows: <520; 520–540; 540–560; 560–580; 580–600; and >600 mm/yr, respectively (Figure 2h).

Vegetation plays a positive role in the stability of landslides and can improve the shear strength of the soil, while increasing the stability of the slope [58]. According to the observations of extensive field investigation, the more vegetation there is, the lower the number of landslides. In light of this, the value of the NDVI, which reflects the degree of vegetation coverage, was reclassified into four ranges based on the natural break method as follows: −0.9315–0.0776; 0.0776–0.4087; 0.4087–0.5742; and 0.5742–2.8915, respectively (Figure 2i).

The slope stability can be influenced by the shape of the slope, which can be evaluated by its profile curvature and plan curvature [59]. The profile curvature is defined as the curvature of a contour line generated by the intersection of the vertical plane with the surface, whereas the plan curvature is defined as that with the horizontal plane [60]. In this study, the profile curvature was classified into five ranges using the natural break method: –15.1897 to –1.5337; –1.5337 to –0.4607; –0.4607–0.5146; 0.5146–1.8802; and 1.8802–9.6837, respectively (Figure 2j). Then, a similar method was applied to divide the plan curvature into five ranges: –9.7777 to –1.8107; –1.8107 to –0.5629; –0.5629–0.3009; 0.3009–1.2608; and 1.2608–14.6991, respectively (Figure 2k).

TWI is commonly used to reflect the water condition in soil [61]. The value of TWI was calculated through the DEM using Equation (1) and was classified into five sections based on the natural break method as follows: 0.0447–2.7551; 2.7551–12.5128; 12.5128–15.0064; 15.0064–18.8011; and 18.8011–27.6913, respectively (Figure 2l).
(1)$$\mathrm{TWI}=\ln\left(\frac{A}{\tan B}\right)$$
where *A* denotes for the specific catchment’s region, and *B* is the value of slope angle in the study area.

TRI was applied to reflect the fluctuation in the surface and the extent of erosion [62]. In the present research, TRI was calculated using Equation (2) and was classified into five ranges: –4508 to –1874; –1874 to –176; –176–57; 57–2398; and 2398–10,418, based on the natural break method (Figure 2m).
(2)$$\mathrm{TRI}=\frac{1}{\cos B}$$

### 3.2. Preparation of Input Data

#### 3.2.1. Frequency Ratio

The input data required for the classification model used in this study were of the numerical type; however, the slope aspect and lithology are nominal variables, so it was necessary to use frequency ratio (FR) data to assign values for these three predisposing factors. The frequency ratio is defined as the ratio of the area where landslides have occurred to the total study region and is also the ratio of the landslide occurrence probabilities to the non-landslide occurrence for a given attribute [63]. The FR data can be calculated according to the following formula:(3)FR=XX′YY′
where *X* and *Y* are the number of landslides in a domain for each class and the number of pixels in a domain for each class, respectively. *X′* and *Y′* stand for the number of total landslides and pixels in the study area, respectively.

In the current research, the slope aspect and lithology factors assigned by the FR values and the remaining 11 predisposing factors, with the original numerical data, were defined as dataset_1_, which was used to run the KLR model.

#### 3.2.2. Box-Counting Dimension

The spatial distribution of landslides is commonly considered to be not uniform, but is instead clustered at different scales. The fractal dimension originating from Mandelbrot’s fractal theory is the value that quantitatively measures the degree of spatial clustering of the landslides. There are many techniques to calculate the fractal dimension, such as the slit island method, box-counting method, and the semi-variance method [64]. The first technique applied in the current research was the box-counting method, and it was employed to calculate the box-counting dimension, which could be used as the input data for landslide susceptibility modeling.

The box-counting method is applicable to both point datasets and can also be used for the calculation of fractal dimension in the two-dimensional and three-dimensional space. The principle of this method is to use a square segmentation plane with side length *ɛ* to calculate the number of grids containing landslide points *N*(*ɛ*), then change the value of *ɛ* to re-divide the plane and calculate the number of grids corresponding to the distribution of landslide points to obtain the sequence of landslide point pair (*ɛ*, *N*(*ɛ*)). In the case where the value *ɛ* is reasonable, if the aforementioned sequences satisfy or approximately satisfy Equation (4), the box-counting dimension (*D*_1_) is considered to exist.
(4)$$N(\varepsilon )\propto \varepsilon^{-D_{1}}$$

Through the python circumstance, the values of the box-counting dimension for each predisposing factor were measured and are shown in Table 4. In addition, 13 predisposing factors assigned by the box-counting dimension values were named as dataset_2_, which was used to run the KLR_box-counting_ model.

#### 3.2.3. Correlation Dimension

The second fractal dimension used as input data for landslide susceptibility modeling was the correlation dimension. The correlation dimension reveals the spatial fractal characteristics and regional differences of landslides from the perspective of the distance between the landslide points, and also reflects the degree of fragmentation of the geomorphological types in the study area. The calculation principle of the correlation dimension is to assume that the number of landslide points is *N*, then set a critical value *r*, determine the landslide point pair where the distance is less than *r*, and calculate its proportion in all landslide point pairs (*N*^2^), as shown in the following formula:(5)$$C(r)=\frac{1}{N^{2}}\sum_{i,j=1}^{N}H(r-\left|X_{i}-X_{j}\right|)$$
(6)$$H(x)=\left\{ \begin{array}{c} 0,x>0\\ 1,x<0 \end{array}\right.$$

If *r* is set too large, then all points are less than r and *C*(*r*) = 1. Therefore, the value of r is gradually increased and the corresponding *C*(*r*) is calculated to obtain a set of sequences. If the above sequences satisfy or approximately satisfy Equation (7), the correlation dimension (*D*_2_) is considered to exist.
(7)$$C(r)\propto r^{-D_{2}}$$

Similarly, the values of box-counting dimension for each predisposing factor were measured based on the python circumstance (Table 4). A total of 13 predisposing factors, assigned by the correlation dimension values, were named as dataset_3_, which was used to run the KLR_correlation_ model.

### 3.3. Multicollinearity Diagnosis

The premise of establishing a regression model is that each explanatory variable is independent of each other. If there is a strong linear correlation between the explanatory variables, it is considered that there is a multicollinearity problem among predisposing factors. The multicollinearity problem may lead to instability in the calculation of regression parameters, which will cause a major error in the results [65]. For these reasons, it is necessary to detect the potential multicollinearity problem between factors. In this study, two indicators obtained from the linear regression analysis, namely variance inflation factors (VIF) and tolerance (TOL), were employed to detect the potential multicollinearity problem. The VIF > 4 or TOL < 0.25 indicates a multicollinearity problem [66]. 

### 3.4. Selection of Predisposing Factors

In the process of landslide susceptibility modeling, not all predisposing factors have a positive influence on the accuracy of the classification modeling. In order to obtain a more accurate and reliable classification result, all of the predisposing factors needed to be filtered by estimating their contribution to the classification model [67]. For this reason, by calculating the information gain ratio (IG) of each predisposing factor to complete the filter process in this study, and the factors whose values of information gain ratio that are equal to or approximately equal to 0 must be excluded before landslide susceptibility modeling. The information gain ratio can be calculated using the following formulas:(8)$$Entropy(D)=-\sum_{k=1}^{\left|y\right|}p_{k}\log_{2}^{p_{k}}$$
where *D* is the training dataset; *Entropy*(*D*) denotes the entropy of the training dataset; and *y* stands for the number of species in *D*. *p_k_* represents the proportion of category *k* in *D*. Then, the training dataset was divided into *D_v_* (*v* = 1, 2, 3, …, m) using *s*, which represents one of the predisposing factors, and we calculated the *Gain*(*D*, *s*) using Equation (9).
(9)$$Gain(D,s)=Entropy(D)-\sum_{v=1}^{\left|m\right|}\frac{\left|D_{v}\right|}{D}Entropy(D_{v})$$

The information gain ratio for predisposing factor *s* is computed as:(10)$$\mathrm{IG}(D,s)=-\frac{Gain(D,s)}{IV(s)}$$
where *IV*(*s*) can be obtained by Equation (11).
(11)$$IV(s)=-\sum_{v=1}^{m}\frac{\left|D_{m}\right|}{\left|D\right|}\log_{2}^{\frac{\left|D_{m}\right|}{\left|D\right|}}$$

### 3.5. Description of the KLR Model

The classification model selected in the current research to construct the landslide susceptibility modeling was a kernel logistic regression model (KLR). KLR is considered as a kernel version of logistic regression [68]. The main principle of the KLR model is to use a kernel function to perform logistic regression operations in high-dimensional feature space on data that are difficult to divide in the current dimensional space [69]. In this study, we took the landslide predisposing factors as input vector *x* and used a kernel function *φ* to complete the non-linear transformation of *x*. Accordingly, the non-linear form of logistic regression can be carried out as follows:(12)logit{p}=w⋅φ(x)+b
where *w* and *b* are preferred by minimizing a cost function to represent the optimal parameters of the model, and *p* is the probability of landslide occurrence. The logit form of Equation (12) can be written as:(13)$$p=\frac{1}{1+\exp\{w\cdot \varphi (x)+b\}}$$

The aforementioned kernel function is defined as the inner product between the images of vectors in the feature space.
(14)K(x,x′)=ϕ(x)⋅ϕ(x′)
There are several kernel functions that have been suggested such as the polynomial kernel, the linear kernel, the radial basis function (RBF), and the sigmoid kernel [70]. In the present research, the kernel function used for modeling was the RBF kernel, which can be written as follows:(15)$$K(x_{i},x_{j})=\exp((-{\left\|x_{i}-x_{j}\right\|}^{2})/2\delta^{2})$$
The kernel sensitivity is controlled by the turning parameter *δ* [71].

### 3.6. Model Evaluation and Comparison

#### 3.6.1. Statistical Index

In this study, the cut-off values were used in the final landslide susceptibility mapping to reclassify the landslide susceptibility index (LSI) into one of the response levels; however, the phenomenon of misclassification always exists in the LSM [72]. In order to evaluate the performance of classification models, six statistical indexes including the positive predictive rate (PPR), negative predictive rate (NPR), sensitivity, specificity, accuracy (ACC), and kappa index were employed as the assessment criteria, and these statistical indexes have frequently been used in many studies [39,73,74]. The PPR, NPR, sensitivity, specificity, and ACC can be calculated based on four basic indexes: the true positive (TP), true negative (TN), false positive (FP), and false negative (FN), as follows:(16)$$\mathrm{PPR}=\frac{\mathrm{TP}}{\mathrm{TP}+\mathrm{FP}}$$
(17)$$\mathrm{NPR}=\frac{\mathrm{TN}}{\mathrm{TN}+\mathrm{FN}}$$
(18)$$\mathrm{Sensitivty}=\frac{\mathrm{TP}}{\mathrm{TP}+\mathrm{FN}}$$
(19)$$\mathrm{Specificity}=\frac{\mathrm{TN}}{\mathrm{TN}+\mathrm{FP}}$$
(20)$$\mathrm{Accuracy}=\frac{\mathrm{TP}+\mathrm{TN}}{\mathrm{TP}+\mathrm{TN}+\mathrm{FP}+\mathrm{FN}}$$
where TP and TN denote the number of pixels which are correctly classified and FN and FP represent the number of pixels which are incorrectly classified.

The kappa index can express the reliability of the classification model, and its calculation process is as follows:(21)$$\mathrm{Kappa}\;\mathrm{index}=\frac{\mathrm{observed}\;\mathrm{accuracy}-\mathrm{chance}\;\mathrm{agreement}}{1-\mathrm{chance}\;\mathrm{agreement}}$$
(22)$$\mathrm{observed}\;\mathrm{accuracy}=\frac{\mathrm{TP}+\mathrm{TN}}{n}$$
(23)$$\mathrm{chance}\;\mathrm{agreement}=\frac{\left(\mathrm{TP}+\mathrm{FN})(\mathrm{TP}+\mathrm{FP})+(\mathrm{FP}+\mathrm{TN})(\mathrm{FN}+\mathrm{TN}\right)}{n^{2}}$$
where n represents the total pixels of the training datasets [75].

#### 3.6.2. The Receiver Operating Characteristic (ROC) Curve

Model comparison is considered as a significant step in landslide susceptibility modeling. In this study, the ROC curve, which is considered to be the most popular and widely used method of comparison models in landslide susceptibility modeling, was applied for assessing the classification model [76]. The *x*-axis and y-axis of the ROC curve are 1-specificity and sensitivity, respectively. The model comparison was undertaken by measuring the value of the area under the ROC curve (AUC), and the calculation formula of AUC is as follows:(24)$$\mathrm{AUC}=\frac{\left(\sum \mathrm{TP}+\sum \mathrm{TN}\right)}{P+N}$$
where P and N denote for the total number of landslides and non-landslides in the study area, respectively.

## 4. Results

### 4.1. Results of Predisposing Factors Analysis

#### 4.1.1. Multicollinearity Diagnosis

In order to detect the potential multicollinearity problems between landslide predisposing factors, the VIF and TOL of dataset_1_, dataset_2_, and dataset_3_ were obtained through linear regression modeling [77]. For dataset_1_, it was observed from Table 2 that the maximum VIF value (1.7055) and the minimum TOL value (0.5863) belonged to the distance to rivers. For dataset_2_, the maximum VIF value (1.2358) and the minimum TOL value (0.8092) belonged to the distance to faults. For dataset_3_, the slope angle had the maximum VIF value and the minimum TOL value, which were 1.2546 and 0.7971, respectively. As a result, the VIF and TOL values of 13 predisposing factors were not within the range of VIF > 4 or TOL < 0.25, indicating that there were no potential multicollinearity problems in dataset_1_, dataset_2_, and dataset_3_.

#### 4.1.2. Predisposing Factors Optimization

In this study, the contribution of predisposing factors for the classification model was quantified by calculating the average merit (AM) as the average IG values using the 10-fold cross-validation. As shown in Table 3, it was obvious that 13 predisposing factors in dataset_2_ and dataset_3_ had a positive contribution to build the classification model (AM > 0). In contrast, the AM values of the TWI, profile curvature, and TRI in dataset_1_ were equal to 0, which means that these three predisposing factors in dataset_1_ had no predictive ability in landslide susceptibility modeling. For this reason, the TWI, profile curvature, and TRI were abandoned from dataset_1_.

### 4.2. Application of the Classification Models

#### 4.2.1. The KLR Model

The FR values of slope aspect and lithology factors and the classification of all predisposing factors are shown in Table 4. The FR value reveals the density of the landslide distribution, and the higher the FR value, the greater the density of the landslide distribution. In the case of slope aspect, the maximum value of FR (1.9024) appeared in the southeast, followed by the south (1.7262), and the east (1.1692), while the minimum FR value was north (0.5845). For lithology, category D had the highest FR value (19.5595), followed by category F with the FR value of 5.0326. 

Dataset_1_ was used as the input data to run the KLR model. The LSI values ranged from 0.0001 to 0.9999. Then, ArcGIS software was applied to visualize the LSI, which should be divided into different ranges to generate the LSM [78]. There are different types of classification schemes such as natural break, quantile, interval, standard deviation, and geometrical interval in ArcGIS software. In the current research, according to the geometrical interval method, the LSI of KLR model was divided into five categories: very low (0.0015–0.2404); low (0.2405–0.3931); moderate (0.3932–0.5615); high (0.5616–0.7494); and very high (0.7495–0.9674). The final LSM of the KLR model is shown in Figure 3a.

#### 4.2.2. Integration of the KLR Model and Fractal Dimension

The acquired box-counting dimensions of dataset_2_ and the correlation dimensions of dataset_3_ are listed in Table 4. For slope angle, the highest box-counting dimension (0.4924) appeared in the section of 18.6711–25.7839°, and the maximum correlation dimension (0.6981) also appeared in this section. In terms of the slope aspect, the class of west had the highest box-counting dimension (0.4469) and correlation dimension (0.6761). As there was no landslide distribution in the class of flat, the two different fractal dimensions were equal to 0. For altitude, the class of 848–1037.6823 m had the highest box-counting dimension and correlation dimension of 0.5758 and 0.7721, respectively. In the case of lithology, the class of category E yielded the maximum box-counting dimension and correlation dimension of 0.9799 and 1.0275, respectively. For distance to roads, the 400–600 m class yielded the highest box-counting dimension and correlation dimension of 0.4974 and 0.7175, followed by the 0–200 m class with a box-counting dimension and correlation dimension of 0.4665 and 0.7005, respectively. In the case of distance to rivers, the maximum box-counting dimension (0.5472) and correlation dimension (0.7544) appeared in the 400–600 m class. For distance to faults, the 0–2000 m class had the highest box-counting dimension and correlation dimension of 0.7235 and 0.8867, respectively. For MAP, the 560–580 mm class yielded the maximum box-counting dimension and correlation dimension of 0.6395 and 0.8342, respectively. In terms of plan curvature, the class of −0.5629–0.3009 had the highest box-counting dimension and correlation dimension of 0.4794 and 0.7002, respectively. For the profile curvature, the maximum box-counting dimension (0.4542) and correlation dimension (0.6857) appeared in the −0.4607–0.5146 class. In the case of TWI, the 2.7551–12.5128 class yielded the highest box-counting dimension and correlation dimension of 0.4653 and 0.6874, respectively. For the NDVI, the 0.5742–2.8915 class yielded the highest box-counting dimension of 0.4694, while the class of 0.0776–0.4087 yielded the maximum correlation dimension of 0.7052. For TRI, the maximum box-counting dimension (0.4859) and correlation dimension (0.7107) appeared in the −176–57 class.

Dataset_2_ was employed as the input data to run the KLR_box-counting_ model. The LSI values of the KLR_box-counting_ model were in the range of 0.0001–0.9999. Then, the LSM of the KLR_box-counting_ model was produced by dividing the LSI values into five categories using the geometrical interval method (Figure 3b). The final threshold segmentation of LSI were as follows: very low (0.0088–0.0610); low (0.0611–0.0765); moderate (0.0766–0.1286); high (0.1287–0.3043); and very high (0.3044–0.9766).

Similarly, dataset_3_ was also employed as the input data to run the KLR_correlation_ model. The LSI values of KLR_correlation_ model were in the range of 0.0001–0.9999. Then, the LSM of the KLR_correlation_ model was produced by dividing the LSI values into five categories using the geometrical interval method (Figure 3c). The final threshold segmentations of LSI were as follows: very low (0.0866–0.3878); low (0.3879–0.5159); moderate (0.5160–0.5704); high (0.5705–0.6986), and very high (0.6987–0.9998).

### 4.3. Model Evaluation

#### 4.3.1. Model Performance

In order to evaluate the performance of the classification models, six statistical indexes including PPR, NPR, sensitivity, specificity, ACC, and kappa index were calculated using the training datasets from dataset_1_, dataset_2_, and dataset_3_. As shown in Table 5, the KLR_correlation_ model yielded the highest PPR, NPR, and ACC of 87.84%, 80.09%, and 83.97%, respectively. For sensitivity, the KLR_correlation_ model showed the best performance for the classification of landslides (sensitivity = 81.59%), followed by the KLR_box-counting_ model (sensitivity = 78.30%), and the KLR model (sensitivity = 66.03%). In terms of specificity, the KLR_box-counting_ model showed the best performance for the classification of non-landslides (specificity = 86.76%), followed by the KLR_correlation_ model (specificity = 87.44%), and the KLR model (sensitivity = 81.67%). Moreover, according to the criteria of the kappa index given from [79]: poor (<0); slight (0–0.2); fair (0.2–0.4); moderate (0.4–0.6); substantial (0.6–0.8); and perfect (0.8–1.0), the KLR_box-counting_ model (kappa index = 0.7657) and the KLR_correlation_ model (kappa index = 0.7828) expressed a substantial reliability. Unfortunately, the KLR model (kappa index = 0.5966) only showed a moderate reliability. 

#### 4.3.2. Model Validation

In this study, the results of model validation using the validation datasets from dataset_1_, dataset_2_, and dataset_3_ are shown in Table 6. The maximum PPR (86.67%), NPR (90.59%), and ACC (88.42%) appeared in the KLR_correlation_ model. For sensitivity, the KLR_correlation_ model expressed the best performance for the classification of landslide (sensitivity = 91.92%), followed by the KLR_box-counting_ model (sensitivity = 83.67%), and the KLR model (sensitivity = 70.41%). For specificity, the KLR_box-counting_ model showed the best performance for the classification of non-landslide (specificity = 85.87%), followed by the KLR_correlation_ model (specificity = 84.62%), and the KLR model (specificity = 79.35%). Furthermore, the kappa indexes of the KLR_box-counting_ model, KLR_correlation_ model, and KLR model were 0.8400, 0.8785, and 0.7336, respectively, indicating a substantial reliability between the reality and models.

### 4.4. Model Comparison

In this study, the model comparison was completed using the AUC value from the ROC curve. Figure 4a shows the final ROC curves and AUC values produced by the training datasets. The KLR_correlation_ model expressed the maximum AUC value of 0.8984, followed by the KLR_box-counting_ model with the AUC value of 0.8828, and the KLR model with the AUC value of 0.8352. 

Additionally, the ROC curves and AUC values produced by the validation datasets are shown in Figure 4b. The KLR_correlation_ model showed the maximum AUC value of 0.9224, followed by the KLR_box-counting_ model with the AUC value of 0.9203, and the KLR model with the AUC value of 0.8605.

## 5. Discussion

The calculated box-counting dimensions and correlation dimensions in this study are listed in Table 4. The value range of the box-counting dimensions was between 0.9261 and 4.6410, while the correlation dimensions ranged from 1.4166 to 6.1590. Although the dimensions of the two fractal methods were different, it can be observed from Figure 5 that the overall trend of variation in the fractal was roughly the same. This indicates that the spatial distribution features of the landslide measured by the two fractal methods were relatively stable and the results more reliable. On the other hand, using the fractal dimension to optimize the predisposing factors may become a new approach that needs to be explored in future research.

Before building the classification models, the potential multicollinearity problems of dataset_1_, dataset_2_, and dataset_3_ were detected. All predisposing factors in these three datasets were independent of each other; however, the difference between dataset_1_, dataset_2_, and dataset_3_ can also be seen from Table 2. In terms of dataset_1_, the TOL values of altitude, distance to roads, distance to rivers, and NDVI were less than 0.7, which seems to indicate that these four factors had a tendency to have multicollinearity problems [80]. Moreover, if these four factors are excluded, it may affect the diversification of the input data. In contrast, the TOL values of all factors in dataset_2_ and dataset_3_ were greater than 0.7, which means that each factor had strong independence as the input data. In addition, from the results of the factor optimization shown in Table 3, three factors including TWI, profile curvature, and TRI in dataset_1_ were excluded, but all predisposing factors in dataset_2_ and dataset_3_ were retained. In summary, dataset_2_ and dataset_3_, which were constructed by the fractal dimension, can maintain a multiplicity of predisposing factors, while dataset_1_ cannot.

The basic classification model used in this study was the KLR model, which is considered as one of the state-of-the art advanced machine learning algorithms [81,82]. Meanwhile the KLR model has been used in landslide susceptibility mapping with high accuracy. However, an exploration of improving the KLR model has seldom been carried out. We used the fractal dimension as the input data of the KLR model for the first time, and the grid search method was applied to ensure that the parameters in the RBF kernel function were optimal at the same time. For model evaluation and comparison, the KLR_correlation_ model constructed by dataset_3_ performed the best, and its AUC values generated by the training dataset and validation dataset were the highest in the three models. Furthermore, the AUC values generated by the KLR model were significantly smaller than the other two models, which may be caused by the excessive difference in the dimension of the original data.

## 6. Conclusions

With the increasing threat of landslides to human beings, the prediction of landslide occurrence is particularly important. Landslide susceptibility mapping is considered as one of the preliminary steps to predict landslide occurrence, the main aim of which is to divide a specified region into multiple classes that range from stable to unstable ones. In this study, to obtain the landslide susceptibility map (LSM), thirteen predisposing factors (i.e., slope aspect, slope angle, altitude, lithology, mean annual precipitation (MAP), distance to rivers, distance to faults, distance to roads, normalized differential vegetation index (NDVI), topographic wetness index (TWI), plan curvature, profile curvature, and terrain roughness index (TRI)) were selected. Then, the KLR model and two hybrid models, namely the KLR_box-counting_ model and the KLR_correlation_ model generated with box-counting dimension and correlation dimension as input data, were used to perform landslide susceptibility mapping in the Baota District, Yan’an City, China.

From the final results, the classification results of all classification models were relatively reliable. For statistical evaluation methods, the performances of the two hybrid models were better than the KLR model. For the result of model comparison, the KLR_correlation_ model had the highest values for landslide susceptibility mapping. 

As the final conclusion, the results in the present study proved that using the fractal dimension as input data to build the hybrid model is feasible for landslide susceptibility mapping in the study area, and could provide a reference for local landslide prevention and decision making. 

## Figures and Tables

**Figure 1 entropy-21-00218-f001:**
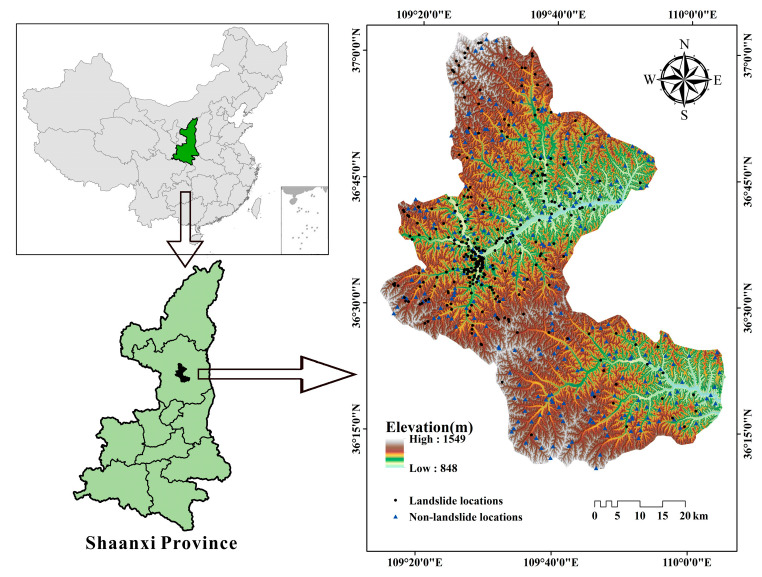
The location of the study area and the landslide inventory map.

**Figure 2 entropy-21-00218-f002:**
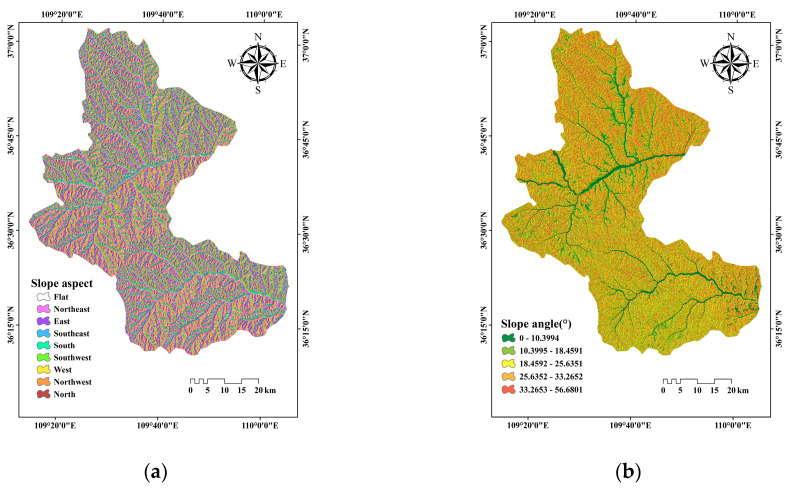
Landslide predisposing factor maps involving: (**a**) Slope aspect; (**b**) Slope angle; (**c**) Altitude; (**d**) Lithology; (**e**) Distance to faults; (**f**) Distance to rivers; (**g**) Distance to roads; (**h**) Mean annual precipitation (MAP); (**i**) Normalized differential vegetation index (NDVI); (**j**) Profile curvature; (**k**) plan curvature; (**l**) Topographic wetness index (TWI); (**m**) Terrain roughness index (TRI).

**Figure 3 entropy-21-00218-f003:**
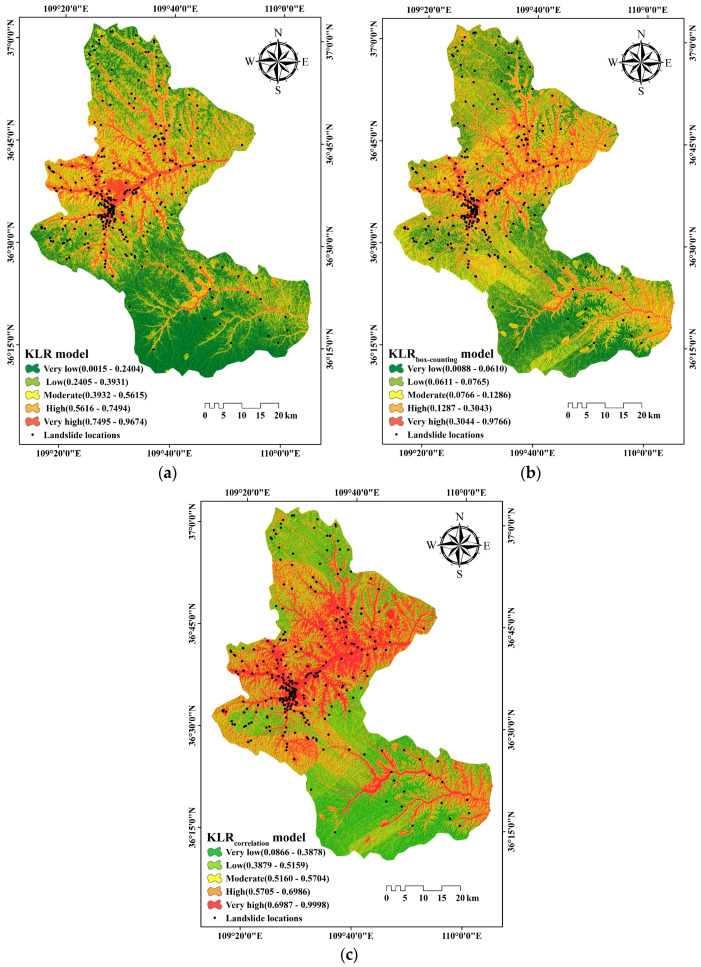
Landslide susceptibility map derived from: (**a**) the kernel logistic regression model (KLR), and; (**b**) the kernel logistic regression based on box-counting dimension model (KLR_box-counting_); and (**c**) the kernel logistic regression based on correlation dimension model (KLR_correlation_).

**Figure 4 entropy-21-00218-f004:**
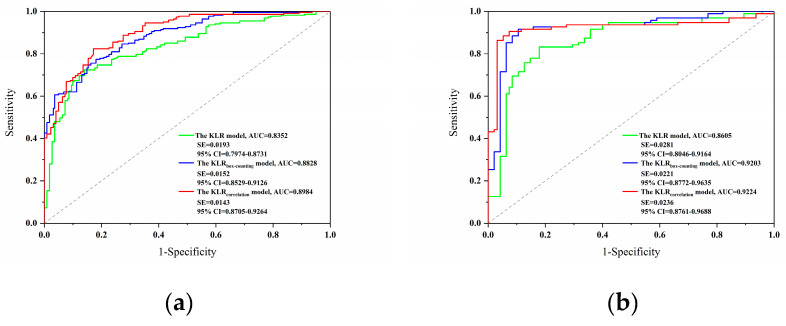
The receiver operating characteristic (ROC) curves of models: (**a**) Training dataset; and (**b**) validation dataset.

**Figure 5 entropy-21-00218-f005:**
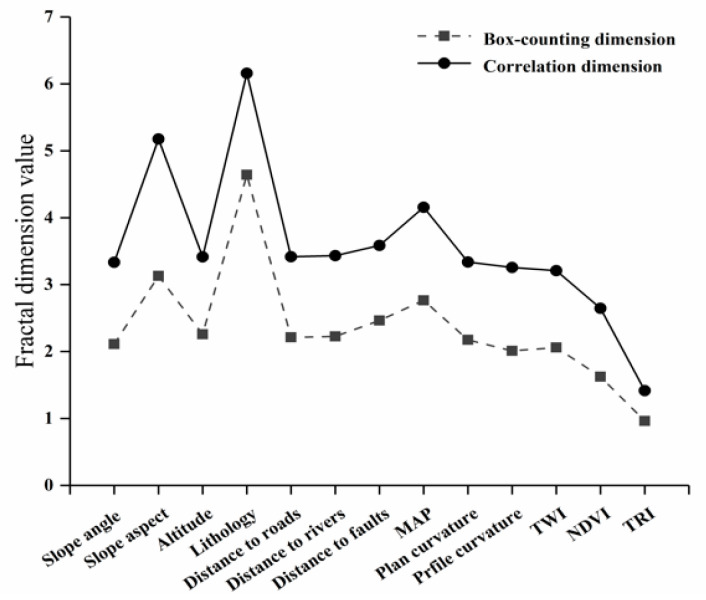
The variation trend of the fractal dimension.

**Table 1 entropy-21-00218-t001:** Lithological units of the study area.

Category	Geological Age	Code	Main Lithology
A	Holocene	Q_h_	Sand and gravel
	Pleistocene	Q_4_	Loess and gravel
	Middle Pleistocene	Q_3_	Loess
B	Pliocene	N_2_	Quartz sand, clay, and sandy clay
C	Early Jurassic	J_3_	Kerosene shale, clumpy conglomerate, glutenite, and silty mudstone
D	Middle Jurassic	J_2_	Arkose, mudstone, and silty mudstone
E	Late Jurassic	J_1_	Sandstone, siltstone, and coal seam
F	Early Triassic	T_3_	Mudstone, shale, and coal seam
G	Middle Triassic	T_2_	Medium-fine sandstone, siltstone, and mudstone
H	Late Triassic	T_1_	Arkose, packsand, siltstone, and sandy mudstone

**Table 2 entropy-21-00218-t002:** The variance inflation factors (VIF) and tolerance (TOL) values of the predisposing factors in the three datasets.

Predisposing Factors	Dataset_1_		Dataset_2_		Dataset_3_	
	VIF	TOL	VIF	TOL	VIF	TOL
Slope aspect	1.0743	0.9308	1.0541	0.9487	1.0784	0.9273
Slope angle	1.2756	0.7839	1.1889	0.8411	1.2546	0.7971
Altitude	1.4321	0.6983	1.1714	0.8537	1.1662	0.8575
Lithology	1.1962	0.8360	1.1842	0.8445	1.1851	0.8438
MAP	1.2652	0.7904	1.1817	0.8462	1.1627	0.8601
Distance to rivers	1.7055	0.5863	1.0322	0.9688	1.0345	0.9667
Distance to faults	1.1681	0.8561	1.2358	0.8092	1.2257	0.8159
Distance to roads	1.5557	0.6428	1.0342	0.9669	1.0433	0.9585
NDVI	1.4661	0.6821	1.0854	0.9213	1.1082	0.9024
TWI	1.0792	0.9266	1.1725	0.8529	1.2246	0.8166
Plan curvature	1.1434	0.8746	1.0552	0.9477	1.0923	0.9155
Profile curvature	1.1812	0.8466	1.0331	0.9680	1.0404	0.9612
TRI	1.0311	0.9698	1.0276	0.9731	1.0301	0.9708

**Table 3 entropy-21-00218-t003:** The information gain ratio (IG) values of predisposing factors in the three datasets.

Predisposing Factors	Dataset_1_		Dataset_2_		Dataset_3_	
	Average Merit	Standard Deviation	Average Merit	Standard Deviation	Average Merit	Standard Deviation
NDVI	0.5111	±0.0072	0.5111	±0.0017	0.5211	±0.0033
MAP	0.4974	±0.0143	0.4731	±0.0214	0.5002	±0.0105
Altitude	0.3865	±0.0111	0.3566	±0.0095	0.3771	±0.0086
Lithology	0.3811	±0.0061	0.3868	±0.0235	0.3588	±0.0059
Distance to roads	0.3806	±0.0047	0.3491	±0.0081	0.3792	±0.0036
Distance to rivers	0.3113	±0.0069	0.3722	±0.0042	0.3643	±0.0024
Slope angle	0.2943	±0.0017	0.3111	±0.0049	0.1016	±0.0075
Distance to faults	0.1295	±0.0095	0.3031	±0.0066	0.3003	±0.0094
Slope aspect	0.1184	±0.0013	0.1002	±0.0054	0.1927	±0.0112
Plan curvature	0.0339	±0.0336	0.1785	±0.0009	0.0922	±0.0058
TWI	0	0	0.2698	±0.0037	0.1047	±0.0044
Profile curvature	0	0	0.0461	±0.0022	0.0705	±0.0021
TRI	0	0	0.0689	±0.0079	0.0553	±0.0083

**Table 4 entropy-21-00218-t004:** The frequency ratio (FR) values and fractal dimensions of each predisposing factor.

Predisposing Factors	Classes	No. of Pixels in Domain	No. of Landslides	FR	Box-Counting Dimension	Correlation Dimension
Slope aspect	Flat	355,630	0	0.0000	0	0
	North	510,563	24	0.5845	0.4408	0.6744
	Northeast	525,473	33	0.7809	0.4056	0.6656
	East	404,148	38	1.1692	0.3383	0.6208
	Southeast	356,144	55	1.9204	0.3603	0.6251
	South	410,618	57	1.7262	0.3762	0.6381
	Southwest	505,082	39	0.9602	0.3738	0.6288
	West	490,901	39	0.9879	0.4469	0.6761
	Northwest	370,883	31	1.0394	0.3871	0.6469
Slope angle (°)	0–10.4469	541,127	75		0.3696	0.6282
	10.4469–18.6711	887,698	103		0.4301	0.6783
	18.6711–25.7839	1,059,498	72		0.4924	0.6981
	25.7839–33.3412	938,160	43		0.4045	0.6498
	33.3412–56.4579	502,959	23		0.4143	0.6793
Altitude (m)	848–1037.6823	519,962	123		0.5758	0.7721
	1037.6823–1128.4000	966,600	105		0.4813	0.7107
	1128.4000–1,210.8706	1,044,874	55		0.3843	0.6338
	1210.8706–1298.8392	902,154	27		0.3971	0.6544
	1298.8392–1549	495,852	6		0.4189	0.6445
Lithology	Category A	2,901,236	139	0.5958	0.8641	0.9942
	Category B	320,975	67	2.5957	0.4914	0.7018
	Category C	34,399	7	2.5304	0.4211	0.6486
	Category D	2543	4	19.5595	0.6594	0.8121
	Category E	25,967	1	0.4789	0.9799	1.0275
	Category F	111,190	45	5.0326	0.4664	0.7044
	Category G	171,799	4	0.2895	0.3386	0.6053
	Category H	361,333	49	1.6863	0.4201	0.6651
MAP (mm/yr)	<520	126,366	3		0.3297	0.6053
	520–540	1,123,449	16		0.3113	0.6053
	540–560	1,376,438	91		0.4277	0.6682
	560–580	771,899	126		0.6395	0.8342
	580–600	457,185	69		0.5926	0.7651
	>600	74,105	11		0.4639	0.6776
Distance to rivers (m)	0–200	238,453	56		0.4583	0.6961
	200–400	235,396	48		0.4044	0.6591
	400–600	231,928	47		0.5472	0.7544
	600–800	228,915	24		0.3627	0.6282
	>800	2,994,750	141		0.4545	0.6942
Distance to roads (m)	0–200	316,529	86		0.4665	0.7005
	200–400	280,765	54		0.4347	0.6894
	400–600	262,675	33		0.4947	0.7175
	600–800	249,049	17		0.3682	0.6235
	>800	2,820,424	126		0.4478	0.6865
Distance to faults (m)	0–2000	689,926	104		0.7235	0.8867
	2000–4000	650,668	68		0.4897	0.7151
	4000–6000	612,815	29		0.4432	0.6797
	6000–8000	510,596	25		0.3906	0.6452
	>8000	1,465,437	90		0.4161	0.6628
NDVI	–0.9315–0.0776	11,230	5		0.3073	0.6053
	0.0776–0.4087	437,324	114		0.4609	0.7052
	0.4087–0.5742	596,564	86		0.3868	0.6411
	0.5742–2.8915	2,885,958	111		0.4694	0.6946
TWI	0.0447–2.7551	1,417,274	87		0.4363	0.6709
	2.7551–12.5128	1,590,117	120		0.4563	0.6874
	12.5128–15.0064	649,808	74		0.4344	0.6745
	15.0064–18.8011	219,949	25		0.3911	0.6428
	18.8011–27.6913	52,294	10		0.3412	0.6053
Plan curvature	–9.7777 to –1.8107	166,235	5		0.4464	0.6584
	−1.8107 to –0.5629	631,367	33		0.4195	0.6645
	–0.5629–0.3009	1,723,931	195		0.4794	0.7002
	0.3009–1.2608	1,087,848	61		0.4112	0.6618
	1.2608–14.6991	320,061	22		0.4182	0.6529
Profile curvature	–15.1897 to –1.5337	293,436	14		0.3789	0.6279
	−1.5337 to –0.4607	905,195	42		0.4409	0.6809
	–0.4607–0.5146	1,650,969	161		0.4542	0.6857
	0.5146–1.8802	827,418	91		0.4205	0.6561
	1.8802–9.6837	252,424	8		0.3154	0.6053
TRI	–4508 to –1874	1,417,271	88		0	0
	–1874 to –176	1,132,853	102		0.4762	0.7059
	–176–57	934,473	80		0.4859	0.7107
	57–2398	361,886	33		0	0
	2398–10,418	82,959	13		0	0

**Table 5 entropy-21-00218-t005:** Model performance using the training datasets.

Statistical Index	Models		
	KLR	KLR_box-counting_	KLR_correlation_
True positive (TP)	173	184	195
True negative (TN)	147	181	177
False positive (FP)	33	26	27
False negative (FN)	89	51	44
Positive predictive rate (PPR) (%)	0.8398	0.8762	0.8784
Negative predictive rate NPR (%)	0.6229	0.7802	0.8009
Accuracy (ACC) (%)	0.7240	0.8258	0.8397
Sensitivity (%)	0.6603	0.7830	0.8159
Specificity (%)	0.8167	0.8744	0.8676
Kappa index	0.5966	0.7657	0.7828

**Table 6 entropy-21-00218-t006:** Model validation using the validation datasets.

Statistical Index	Models		
	KLR	KLR_box-counting_	KLR_correlation_
TP	69	82	91
TN	73	79	77
FP	19	13	14
FN	29	16	8
PPR (%)	0.7841	0.8632	0.8667
NPR (%)	0.7157	0.8316	0.9059
ACC (%)	0.7474	0.8474	0.8842
Sensitivity (%)	0.7041	0.8367	0.9192
Specificity (%)	0.7935	0.8587	0.8462
Kappa index	0.7336	0.8400	0.8785

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
