# Peer review of "Assessment of Landslide Susceptibility Using Integrated Ensemble Fractal Dimension with Kernel Logistic Regression Model"

_entropy, 2019, doi:10.3390/e21020218_

Round 1

Reviewer 1 Report

The article focuses on the evaluation of the performance of fractal dimension as input data in landslide susceptibility analysis. 

The study is based on an inventory of 316 landslide points coupled with 316 non-landslide points located in the central China (Baota district, Yan’an City). 

It considers, within statistical analysis, a comprehensive amount of thirteen factors predisposing landslides triggering.

The paper appears well set up, well developed and models performance and dataset validation are clearly stated.

Just some minor revisions are suggested for the text, the figures and the tables as indicated in the attached file.

Author Response

 Thanks a lot for your comments, please see our responses in the attachment.

Reviewer 2 Report

Main comments

1) too many abreviations and special terms such as shown in the text

2)How did you find the existing landslides? by air photos or Google Earth?

3)this paper is dealing with landslide, but too many descrinptions on technology, so the authors should consult with geologists or geomorpholoigsts concerning about landslides

4) in the Conclusion, describe the significance of the methods for landslide susceptibility mapping. How is a benefit and difference from the other methods for  the landslide susceptibility mapping 

Author Response

Response to Reviewer 2 Comments

Dear reviewer:

I am very grateful to your comments for the manuscript. According with your advice, we amended the relevant part in manuscript.  Your questions were answered below.

Point 1:  Too many abreviations and special terms such as shown in the text.

Response 1: As Reviewer suggested that the language of manuscript has been edited by MDPI English editing office. The English editing certificate is shown below.

Point 2: How did you find the existing landslides? by air photos or Google Earth

Response 2: We are sorry for our negligence. In this work, the total of 316 landslides were identified using existing literatures and reports, field survey data and results of interpretation of aerial photographs.

Point 3:  This paper is dealing with landslide, but too many descrinptions on technology, so the authors should consult with geologists or geomorpholoigsts concerning about landslides.

Response 3: We are sorry about our lack of knowledge in geology and geomorphology. In the further research, we will do our best to consider landslides from the perspective of geology and geomorphology. We describe the type and plane proportion of landslides in the Section 3.1.1, as follows:

There were 316 landslides including four debris flows, 295 rainfall-induced slides, and 17 falls in the landslide inventory map [48], and the largest plane proportion of landslides was approximately 11.4×104 m2, the minimum area was about 295 m2 and the average proportion was 61 m2.

48.          Hungr, O.; Leroueil, S.; Picarelli, L. The varnes classification of landslide types, an update. Landslides 2014, 11, 167-194.

Point 4: In the Conclusion, describe the significance of the methods for landslide susceptibility mapping. How is a benefit and difference from the other methods for the landslide susceptibility mapping.

Response 4: Considering the Reviewr’s suggestion. We describe the significance of the methods for landslide susceptibility mapping, and this part can be seen in Section 5, as follows:

The basic classification model used in this study is the KLR model, which is considered as one of the state-of-the art advanced machine learning algorithm [81,82]. Meanwhile the KLR model has been used in landslide susceptibility mapping with high accuracy. However, an exploration of improving the KLR model has seldom been carried out. We used the fractal dimension as the input data of the KLR model for the first time, and the grid search method was applied to ensure that the parameters in the RBF kernel function were optimal at the same time. For model evaluation and comparison, the KLRcorrelation model constructed by the dataset3 performs best, and its AUC values generated by the training dataset and validation dataset are the highest in the three models. Furthermore, the AUC values generated by the KLR model are significantly smaller than the other two models. Therefore, the KLRcorrelation model is consider as the most suitable classification model for the study area.

81.          Chen, W.; Shahabi, H.; Zhang, S.; Khosravi, K.; Shirzadi, A.; Chapi, K.; Pham, B.T.; Zhang, T.; Zhang, L.; Chai, H., et al. Landslide susceptibility modeling based on gis and novel bagging-based kernel logistic regression. applied sciences 2018, 8, 2540.

82.          Hong, H.; Pradhan, B.; Xu, C.; Bui, D.T. Spatial prediction of landslide hazard at the yihuang area (china) using two-class kernel logistic regression, alternating decision tree and support vector machines. Catena 2015, 133, 266-281.

Round 2

Reviewer 2 Report

This manuscript is improved well for publication